# Efficient Dual-Confounding Eliminating for Weakly-supervised Temporal Action Localization

## ABSTRACT

Weakly-supervised Temporal Action Localization (WTAL) following a localization-by-classification paradigm has achieved significant results, yet still grapples with confounding arising from ambiguous snippets. Previous works have attempted to distinguish these ambiguous snippets from action snippets without investigating the underlying causes of their formation, thus failing to effectively eliminate the bias on both action-context and action-content. In this paper, we revisit WTAL from the perspective of structural causal model to identify the true origins of confounding, and propose an efficient dual-confounding eliminating framework to alleviate these biases. Specifically, we construct a Substituted Confounder Set (SCS) to eliminate the confounding bias on action-content by leveraging the modal disparity between RGB and FLOW. Then, a Multi-level Consistency Mining (MCM) method is designed to mitigate the confounding bias on action-content by utilizing the consistency between discriminative snippets and corresponding proposals at both the feature and label levels. Notably, SCS and MCM could be seamlessly integrated into any two-stream models without additional parameters by Expectation-Maximization (EM) algorithm. Extensive experiments on two challenging benchmarks including THUMOS14 and ActivityNet-1.2 demonstrate the superior performance of our method.

## KEYWORDS

Weakly-supervised, Temporal Action Localization, Structural Causal Model, Substituted Confounder Set, Consistency Mining.

**ACM Reference Format:**
Anonymous Author(s). 2024. Efficient Dual-Confounding Eliminating for Weakly-supervised Temporal Action Localization. In *Proceedings of Proceedings of the 32th ACM International Conference on Multimedia (MM '24)*. ACM, New York, NY, USA, 10 pages. https://doi.org/XXXXXXX.XXXXXXX

## 1 INTRODUCTION

Temporal action localization (TAL) refers to the task of localizing the start and end timestamps of action proposals and identifying their labels in untrimmed videos, which is crucial in a wide variety of video understanding applications [45]. Typically, fully-supervised temporal action localization [41, 22, 51, 44, 25, 16] has achieved significant localization results. However, fully-supervised methods rely on a huge amount of expensive fine-grained frame-level

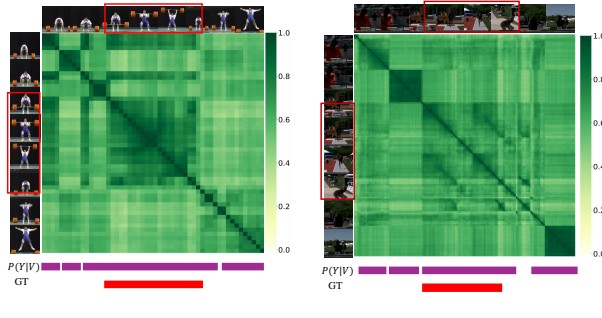

**Figure 1: Confounding bias on action-context.** $P(Y|V)$ stands the proposals generated by the previous WTAL models. (a) Visual confounding in "CleanAndJerk": visual similarity as a confounder introduces the false activations for action-context. (b) Motion confounding in "LongJump": imperceptible distinctions of motion speed among pre-jump, jump, and post-jump phases brings up the same confounding issue.

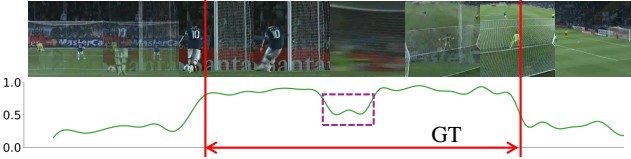

**Figure 2: Confounding bias on action-content.** Existing WTAL model $P(Y|V)$ is confused when variability and individualization occur within action-content, which is mainly reflected in the ambiguous attention scores for less discriminative content snippets.

annotations. To address this issue, weakly-supervised temporal action localization (WTAL) [59, 35, 7, 52, 21, 54, 37, 36] with only video-level annotations has recently gained intensive attention.

Most WTAL approaches tackle the annotation sparsity challenge by reconceptualizing localization as a classification task to identify temporal regions most relevant to video-level classification [59, 35, 7, 52]. While these methods have achieved certain performances, they still suffer from confounding [26] arising from ambiguous snippets, leading to suboptimal results. Existing approaches usually categorize these ambiguous snippets under a newly devised "action-context" label, striving to segregate them from definitive action snippets. For instance, ACSNet [27] first brings in the assistant category "context" to assist in separating foreground action from context, while FACNet[11] utilizes class-wise foreground classification branch to regularize the relation between actions and foreground. Similarly, DDG-Net[43] designs a graph network to

explicitly model ambiguous snippets. However, these methods often oversimplify ambiguous snippets into a singular, predefined category without investigating the underlying origins of their formation, consequently failing to eliminate the confounding bias efficiently.

In light of these challenges, it is crucial to identify the underlying causes of confounding bias and mitigate them. Inspired by [60, 38], we revisit WTAL from the perspective of structural causal model [4]. Specifically, existing WTAL models predict action label $y_{ij}$ for every video snippet $v_{ij}$ in video $V_i$ via a classification model $P(Y|V)$. The scarcity of snippet-specific annotations causes models to excessively focus on most discriminative action snippets, thereby introducing biases on both action-context and action-content. More specifically, action-context owning visual or motion similarities with discriminative action snippets(as illustrated in Fig. 1(a) and Fig. 1(b)) may erroneously be classified as action instances. When action-content exhibit some uniqueness diverging from the majority of discriminative action snippets, $P(Y|V)$ is inclined to assign them ambiguous confidence values, as shown in Fig. 2. Generally, these bias on action-context and action-content is a significant challenge that has yet to be effectively addressed.

Fortunately, structural causal model offers a solution through $P(Y|do(V))$ instead of $P(Y|V)$ to eliminate the confounding bias. The $do(\cdot)$ operation [4] denotes the causality between the cause $V$ and the effect $Y$ without confounders[34]. The ideal way to calculate $P(Y|do(V))$ is to "physically" intervene $V$, such as removing all ambiguous snippets in every action instance. Obviously, it is impossible to intervene the video snippets in practice.

To address these challenges, we propose an efficient dual-confounding eliminating framework to mitigate the confounding bias on action-context and action-content through two core components, respectively. The first component, termed Substituted Confounder Set (SCS) is constructed by leveraging the modal disparity between RGB and optical FLOW. Then, SCS is applied in a backdoor adjustment procedure [34] to address context-related bias. Unlike previous methods [43, 26], the construction of SCS does not require an elaborate network architecture or additional learnable parameters, making it more efficient and effective. The second component, named Multi-level Consistency Mining (MCM), is designed to tackle action-content bias by ensuring consistency between discriminative snippets and respective proposals on both feature and label levels. Different from previous works [12, 65, 14, 19] selecting the representative action embeddings across the entire dataset, yet disregarding the individualized differences among specific action instances, MCM selects the representative snippets from corresponding proposals for each action instance, to mitigate the action-content confounding on these less discriminative snippets by enforcing the model to learn the consistency across each action instance. Ultimately, by integrating the current WTAL models with the SCS and MCM components, we can effectively approximate $P(Y|do(V))$ using the Expectation-Maximization (EM) algorithm, while maintaining low computational costs.

In summary, our main contributions are summarized as follows: (1) We revisit WTAL from the perspective of structural causal model, and propose an efficient dual-confounding eliminating framework to mitigate the confounding bias. (2) We construct a Substituted Confounder Set (SCS) to specifically eliminate the confounding

bias on action-context by exploiting the modal disparity between RGB and FLOW. (3) We present the Multi-level Consistency Mining (MCM) method to address the action-content bias, utilizing consistency between discriminative snippets and associated proposals on both feature and label levels. (4)The SCS and MCM could be applied to any two-stream model without additional learnable parameters using EM algorithm. Extensive experiments on THUMOS14 [13] and ActivityNet-1.2 [9] demonstrate the superior performance of our methods.

## 2 RELATED WORK

**Weakly-supervised Temporal Action Localization.** To solve the issue of fine-grained annotations, weakly-supervised techniques[59, 7, 35, 54, 52, 37, 36, 21] have emerged, enabling the localization of action instances with only video-level labels. Among these, UntrimmedNet[47] stands out as a pioneering effort, employing a Multiple Instance Learning (MIL) framework [29] to tackle the weakly-supervised temporal action localization task. Attention-based methods [10, 31] have also gained traction, offering enhanced performance and architectural flexibility. These methods, including BaS-Net[17] and WSAL-BM[32], could be used to suppress activations from background frames. More recently, pseudo labels are used to supervise the model to generate more accurate attention scores for video snippets[53]. TSCN[59] fuses RGB and FLOW modalities to generate better pseudo labels, while work[54] considers the complementarity. Traditionally, RGB and FLOW modalities have been treated uniformly, with their concatenated features at the feature level providing a straightforward yet limited approach to exploiting modality distinctions, such as ASM-Loc[8], RSKP[12], GauFuse[65], TFE-DCN[64], P-MIL[36] and so on, but it is not sufficient to capture the subtle differences between RGB and FLOW modalities. In contrast, another research direction focuses on calculating separate attention scores for each modality before combining them, aiming to better harness their complementary aspects, liking $CO_2$-Net[10], DELU[3], DDG-Net[43] and so on. However, these methods are not fully utilize collaborations between modalities. In this paper, we mitigate confounding bias by exploiting the unique differences and synergies between RGB and FLOW modalities.

**Causal Intervention and Deconfounder.** Causal intervention has been recently introduced to address the confounding bias in computer vision tasks, including image semantic segmentation[60], image captioning[55], image classification[42, 58], object detection [38] and so on. This bias presents challenges not just in image-based tasks but also in video analyses, particularly spatial-temporally complex videos, where video temporal grounding [30, 48], video action localization [26], video action anticipation [61], moment retrieval [56] etc. are explored. For example, work [60] propose a structural causal model that analyzes the causal relationships among images, contexts, and class labels. Based on this model, they develop a novel method called Context Adjustment (CONTA). DCM [56] introduces a causal model to discern the direct influence of queries and video content on outcomes, sidestepping the confounding effects of moment locations by isolating the core visual content features. For WTAL, Liu et al[26] point out that the "background" issue is impossible to be fundamentally resolved due to weak supervision, but they simply attribute every spurious association to an unobserved

confounding and propose to learn a substitute by adopting PCA on temporal dimension. However, this approach rests on stringent assumptions, limiting its applicability. Different from previous works that consider the confounding bias on whole video level and model the confounding with additional parameters or extra storage costs, we seek to efficient eliminate the confounding bias on both action-context and action-content separately with the consideration of individualized differences among specific action instances.

## 3 THE PROPOSED METHOD

In this section, we detail our efficient dual-confounding elimination framework. The overview of the proposed methods is shown in Fig. 3, the framework follows the EM algorithm to iteratively mitigate confounding bias. Initially, we give the task formation (Section 3.1). Subsequently, we revisit WTAL from the perspective of structural causal model (Section 3.2). Following this, we construct the Substituted Confounder Set (SCS) to approximate action-context confounder (Section 3.3). To eliminate action-content confounding bias, we propose the Multi-level Consistency Mining (MCM) method in Section 3.4. Notably, our method can be applied to any two-stream models [10, 3, 43] without any additional learnable parameters by the EM algorithm. Finally, we introduce the training and inference details in Section 3.5.

### 3.1 Task Formulation

WTAL aims to identify action instances within untrimmed videos under the guidance of only video-level labels. Suppose we have a dataset of training videos represented as $\{\mathbf{F}_i\}_{i=1}^{N}$ is given, where $N$ is the number of training videos and $\mathbf{F}_i \in \mathbb{R}^{T \times D}$ is the extracted $D$-dimensional video features of $T$ segments. Only the corresponding video-level label $\mathbf{y}_i \in \{0, 1\}^M$ is available to train models for each video $\mathbf{V}_i$, where $M$ is the number of classes. During testing, the goal is to get a set of action proposals $S = (s_i, e_i, c_j, \varphi_i)$ for each testing video, where $s_i$ and $e_i$ are the start and the end times of an action instance respectively, $c_j$ indicates the predicted action class, and $\varphi_i$ denotes the related confidence score.

### 3.2 Structural Causal Model for WTAL

In this section, we revisit WTAL with the perspective of structural causal model [4]. As illustrated in Fig. 4(b), our structural causal model encapsulates four key variables: action prior $C$, video $V$, action label $Y$ and the diverse representations of action instances $I$. The model establishes direct causal links, indicating cause-and-effect relationships between pairs of variables: cause → effect. Conventional WTAL methods (as depicted in Fig. 4(a)) typically focus on learning from two main causal links: $V \rightarrow Y$ and $C \rightarrow Y$ to improve the model's generalization, but they overlook the variability among different action instances within the same action class. While our methodology scrutinizes the confounding bias at the instance-level, which is crucial for uncovering the underlying causes of the confounding bias.

$C \rightarrow V$. Action prior $C$ derived from the entire dataset and considered as a universal action templates, influences the action depicted in video $V$. $C$ not only tells the model what to expect in video $V$ and how to act in the video, which is specifically reflected in the RGB modality and FLOW modality, but also contains the action

context that may lead the model to mistakenly identify snippets with similar visual appearances or motion patterns as actual actions. It is a significant challenge to accurately represent the $C \rightarrow V$ relationship in complex action scenarios. Fortunately, we can avoid the action-context bias through SCC, which will be introduced in Section 3.3.

$V \rightarrow I \leftarrow C$. $I$ represents the distinct action instance, derived by the action templates from $C$. $V \rightarrow I$ indicates the causal relationship between video $V$ and action instances $I$, while $C \rightarrow I$ depicts that instances are also determined by action prior $C$. For example, the action "Diving" may occur many times in a video $V_i$, but not all instances follow the same procedure, any "Diving" action instance $I_{ij}$ can be descried by "Diving" template $C_{Diving}$. Importantly, there exists both commonalities and individual characteristics within different action instances of the same action category. Previous methods in Fig. 4(a) usually focus on the commonalities, the overlook of instance-specific individualities makes models not confident for the less discriminative content snippets, resulting in action-content bias. Our perspective in Fig. 4(b) is to balance the commonalities and individual characteristics.

$V \rightarrow Y \leftarrow I$. In WTAL task, video class labels $Y$ typically come from the conventional classification model $V \rightarrow Y$ ($P(Y|V)$). In fact, $Y$ is also influenced by the mediation $I$. $I \rightarrow Y$ denotes an obvious causality that the action instances within a video determine the video-level class label. It is worth noting that the action instances $I$ are not used to supervise the model in WTAL task. The relationship $C \rightarrow Y$ is not shown in the structural causal model, because the action prior template $C$ indirectly affects $Y$ through the action instances $I$. If $I \rightarrow Y$ does not exist, there is an only path $C \rightarrow V \rightarrow Y$ left from $C$ to $Y$, which is impossible for action prior $C$ to make effect to $Y$, and we would never uncover the confounding bias in WTAL. Multi action categories and multi action instances within a video prove the necessity of $I \rightarrow Y$.

So far, we have established the structural causal model for WTAL task, showcasing it as an instrumental framework for uncovering the genuine causal connections among variables. Thanks to the graphical representation, we can clearly see how $C$ confounds $V$ and $Y$ via the backdoor path $V \leftarrow C \rightarrow I \rightarrow Y$, even if some video snippets in $V$ have nothing to do with $Y$, the backdoor path can still help to correlate them with $Y$, resulting the confounding bias. However, $C$ and $I$ remain unobservable and immeasurable in practice, posing challenges in directly intervening the backdoor path. Nonetheless, constructing a substitute for the unobserved confounder on $C$ is feasible by leveraging the modal disparity between RGB and FLOW modalities, and the substitute could be used to intervene the video snippets $V$ to eliminate action-context confounding. Similarly, the confounding bias related to the action-content instances $I$ could be mitigated by enforcing model to learn the consistency across each action instance. Ultimately, $P(Y|do(V))$ could be approximated.

### 3.3 Substituted Confounder Set

The existing of backdoor path $C \rightarrow V$ in the structural causal model leads to the confounding bias. As "physical" intervention is impossible, we apply the backdoor adjustment procedure [34] to achieve $P(Y|do(V))$. To clearly illustrate the process, we stratify $C$ into pieces $C = \{c_j \in C\}$, where $c_j$ represents the action prior for

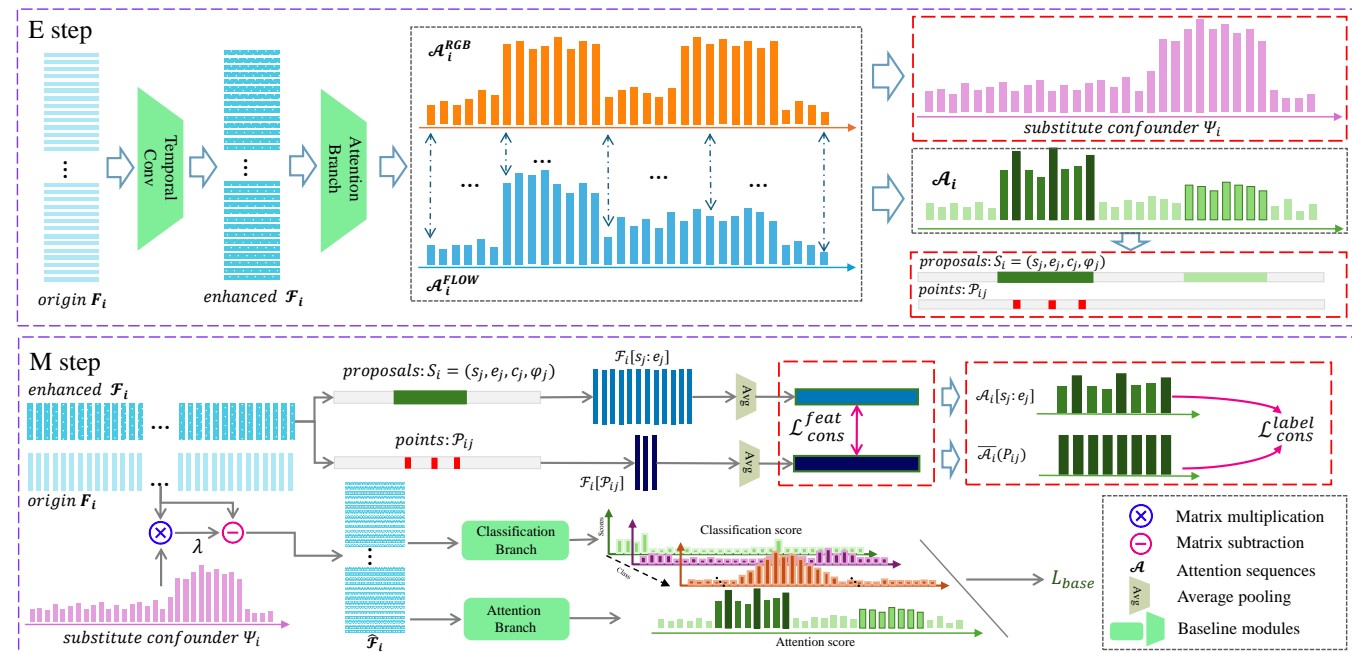

Figure 3: Overview of the dual-confounding eliminating framework, following EM algorithm to iteratively eliminate the confounding bias on action-context and action-content. At E step, $\mathcal{A}_i^{RGB}$ and $\mathcal{A}_i^{FLOW}$ are calculated by existing WTAL models, and then the substituted confounder set $\Psi_i$ is constructed by Eq. (2) to approximate the confounding of action-context. Action proposals and discriminative snippets are generated by threshold $\theta_{pro}$ and $\theta_{rep}$ respectively. At M step, $do(\cdot)$ is realized by suppressing $\Psi_i$ on original video features $F_i$. Besides, the consistency constraints between discriminative snippets and corresponding proposals are enforced on both feature-level and label-level to mitigate the action-content bias.

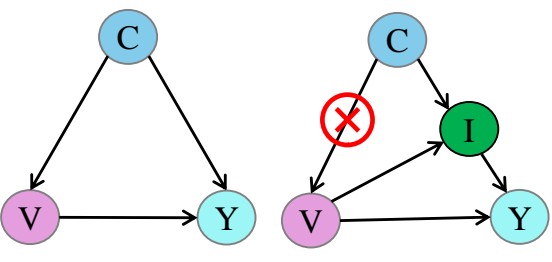

(a) Conventional SCM for WTAL   (b) Proposed SCM for WTAL

Figure 4: WTAL under the perspective of structural causal model (SCM). Conventional SCM presented in (a) just considers the causalities among video $V$, action prior $C$ and action label $Y$ without considering the individual differences among action instances. We revisit conventional WTAL in (b) considering the causalities among video $V$, action label $Y$, action prior $C$ and different action instance representation $I$, which is more comprehensive for identifying the true origins of confounding bias.

category $i$. The causal intervention can be formulated as:

$$P(Y|do(V)) = \sum_{c_j \in C} P(Y|V, I)P(c_j) \qquad (1)$$

where $I = f(V, c_j)$ signifies the action instance within the localization task, $f(\cdot)$ is the function that generates instances $I$ from $V$ and $C$. $P(Y|V, I)$ represents the conditional probability distribution that WTAL models need to learn. According to Eq. (1), $C$ is no longer correlated with $V$, the causal intervention makes $V$ have a fair opportunity to incorporate every context $c_j$ into the prediction of $Y$ through $P(c_j)$. Now the challenge is how to obtain the unobserved and unmeasurable confounder on $C$ in practice.

Fortunately, based on the discussion in Section 1 that visual ambiguity and motion ambiguity tend not to occur simultaneously [43], the modal disparity between RGB and FLOW could be utilized to construct a substitute for the confounder on $C$. For a video $V_i$, we can get $\mathcal{A}_i^{RGB}$ and $\mathcal{A}_i^{FLOW}$ respectively through existing WTAL model. Then the substitute confounder $\Psi_i$ could be constructed by

$$\Psi_i = Index\left(\frac{|\mathcal{A}_i^{RGB} - \mathcal{A}_i^{FLOW}|}{|\mathcal{A}_i^{RGB} + \mathcal{A}_i^{FLOW}|} > \delta\right) \qquad (2)$$

in which $Index$ is indicator function to get the index of the condition, $\delta$ is a threshold to measure the gap between RGB and FLOW modalities. After getting $\Psi_i$, we can get the representation of video $V_i$ with causal intervention by

$$\hat{\mathcal{F}}_i = \mathcal{F}_i - \lambda \Psi_i \mathcal{F}_i \qquad (3)$$

where $\mathcal{F}_i$ is the representation of $V_i$, $\lambda$ is the weight to control the strength of the intervention.

The whole process of our causal intervention for video $V_i$ can be formulated as:

$$
\begin{aligned}
P(Y_i|do(V_i);\Theta) &= \sum_{c_j \in C} P(Y_i|V_i, I_i;\Theta)P(c_j;\Theta) \\
&\approx \sum_{c_j \in C} P(Y_i|\mathcal{F}_i, I_i;\Theta)P(c_j;\Theta) \\
&\approx \sum_{c_j \in C} P(Y_i|\hat{\mathcal{F}}_i, I_i;\Theta)P(c_j;\Theta)
\end{aligned}
\tag{4}
$$

where $\Theta$ is the parameters of the WTAL model, and $P(Y|\hat{\mathcal{F}}_i, I_i;\Theta)$ is the prediction of the WTAL model after the causal intervention. By constructing the substitute confounder $\Psi_i$ and intervening the video feature $\mathcal{F}_i$, we could effectively eliminate the confounding bias on action-context.

### 3.4 Multi-level Consistency Mining

Regarding the action-content bias, it arises because the classification loss steers the model towards concentrating on the most discriminative snippets. This focus may lead to the potential neglect of less discriminative action-content snippets, which are also crucial for accurate model performance. To address this issue, we propose a Multi-level Consistency Mining method to leverage the consistency between discriminative snippets and corresponding proposals on feature-level and label-level to effectively mitigate the action-content confounding.

For feature-level consistency mining, instead of previous works [12] that select the representative action embeddings across the entire dataset while disregarding the individuality among same category action instances, we propose to pick up the representative snippets from corresponding proposals for each action instance, our approach then minimizes the distance between the representative action embeddings and the corresponding proposals. Specifically, following the previous work [10, 3, 43], we can get the final attention scores $\mathcal{A}_i$ through existing WTAL model for video $V_i$, and then proposal set $S_i = (s_j, e_j, c_j, \varphi_j)$ can be generated by threshold $\theta_{pro}$. For each proposal $s_j, e_j$ in $S_i$, we first select the most representative snippets $\mathcal{P}_{ij}$ by:

$$
\mathcal{P}_{ij} = Index(\mathcal{A}_i[s_j : e_j] > \theta_{rep})
\tag{5}
$$

where $\theta_{rep}$ is the threshold to measure the representative snippets. Then, corresponding representative snippets feature can be calculated by:

$$
\mathcal{F}_{ij}^{rep} = \mathcal{F}_i[\mathcal{P}_{ij}]
\tag{6}
$$

The feature of corresponding proposal $s_j, e_j$ can be calculated by:

$$
\mathcal{F}_{ij}^{pro} = \mathcal{F}_i[s_j : e_j]
\tag{7}
$$

We aim to minimize the distance between the representative action embeddings and the corresponding proposals, and the feature-level consistency loss can be formulated as follows:

$$
\mathcal{L}_{cons}^{feat} = \sum_{i=1}^{N} \sum_{j=1}^{K} \left\| avg(\mathcal{F}_{ij}^{rep}) - avg(\mathcal{F}_{ij}^{pro}) \right\|_2
\tag{8}
$$

where $N$ is the number of training videos, $K$ is the number of proposals in video $V_i$.

The consistency in features for each action instance should accompany the consistency in labels[49], we consider that the consistency constraints of attention scores for proposals enable the model to treat the action instances within the same category more equitable and could further eliminate the action-content confounding bias. More specifically, after getting proposal set $S_i = (s_j, e_j, c_j, \varphi_j)$ and representative snippets $\mathcal{P}_{ij}$ for video $V_i$, for each proposal $(s_j, e_j, c_j, \varphi_j)$, the attention score sequences of $(s_j, e_j, c_j, \varphi_j)$ can be got by:

$$
\mathcal{A}_{ij}^{pro} = \mathcal{A}_i[s_j : e_j]
\tag{9}
$$

The average attention score of the representative snippets $\mathcal{P}_{ij}$ can be calculated by:

$$
\bar{\mathcal{A}}_{ij}^{rep} = avg(\mathcal{A}_i[\mathcal{P}_{ij}])
\tag{10}
$$

Then, we use $\bar{\mathcal{A}}_{ij}^{rep}$ to supervise all snippets in the proposal $(s_j, e_j, c_j, \varphi_j)$. The label-level consistency loss can be formulated as:

$$
\mathcal{L}_{cons}^{label} = \sum_{i=1}^{N} \sum_{j=1}^{M} \sum_{t=s_j}^{e_j} \left\| \bar{\mathcal{A}}_{ij}^{rep} - \mathcal{A}_i[t] \right\|_2
\tag{11}
$$

Our final multi-level consistency mining loss function is expressed by integrating all the above constraints.

$$
\mathcal{L}_{cons} = \beta \mathcal{L}_{cons}^{feat} + \gamma \mathcal{L}_{cons}^{label}
\tag{12}
$$

where $\beta$ and $\gamma$ are the weights to control the strength of the feature-level and label-level consistency mining respectively.

### 3.5 Model Training and Inference

**Optimizing Process**. The overall training objective of the proposed method can be formulated as:

$$
\begin{aligned}
\mathcal{L}_{total} &= \mathcal{L}_{base} + \mathcal{L}_{cons} \\
&= \mathcal{L}_{base} + \beta \mathcal{L}_{cons}^{fea} + \gamma \mathcal{L}_{cons}^{lab}
\end{aligned}
\tag{13}
$$

where $\mathcal{L}_{base}$ is the train objectives of the baseline WTAL models, which including $CO_2$-Net[10], DELU[3], DDG-Net[43]. More details about the experiments and results will be introduced in the following section.

**Model Inference**. In the testing phase, we adhere to the same pipeline as the baseline WTAL models [10, 3, 43] to generate action proposals, and we employ the same test settings and post-processing methods.

## 4 EXPERIMENTS

### 4.1 Datasets

**THUMOS14.** THUMOS14[13] dataset comprises 200 validation videos and 213 test videos that offer frame-wise annotations for 20 classes, and with an average of 15.5 actions per video. We train the model on the untrimmed videos in its validation set and evaluate it on the untrimmed videos from the test set following the same setting as [10, 3, 43].

**ActivityNet-1.2**: ActivityNet-1.2 [9] dataset covers 200 daily activities and provides 10,024 videos for training, 4,926 for validation and 5,044 for testing offering a larger benchmark for temporal action localization. We use all the training set to train model and all the validation set to evaluate our methodologies.

**Table 1: The proposed methods are applied to different WTAL methods on THUMOS14 dataset. "Abs. Improve" denotes the absolute improvement of our method over the baseline methods**

| Method | mAP@IoU(%) | | | | | | | AVG | | |
|---|---|---|---|---|---|---|---|---|---|---|
| | 0.1 | 0.2 | 0.3 | 0.4 | 0.5 | 0.6 | 0.7 | (0.1:0.5) | (0.3:0.7) | (0.1:0.7) |
| $CO_2$-Net(MM'21)[10] | 70.1 | 63.6 | 54.5 | 45.7 | 38.3 | 26.4 | 13.4 | 54.4 | 35.7 | 44.6 |
| $CO_2$-Net$^\dagger$ (MM'21)[10] | 71.8 | 64.9 | 56.4 | 45.8 | 34.1 | 23.1 | 13.2 | 54.6 | 34.5 | 44.2 |
| **Ours** | 73.1 | 66.9 | 58.7 | 48.9 | 37.2 | 24.5 | 15.5 | 56.9 | 37.0 | 46.4 |
| Abs. Improve | 1.3 | 2.0 | 2.3 | 3.1 | 3.1 | 1.4 | 2.3 | 2.3 | 2.5 | 2.2 |
| DELU(ECCV'22)[3] | 71.5 | 66.2 | 56.5 | 47.7 | 40.5 | 27.2 | 15.3 | 56.5 | 37.4 | 46.4 |
| DELU$^\dagger$(ECCV'22)[3] | 71.3 | 66.2 | 56.8 | 47.2 | 40.1 | 27.5 | 14.9 | 56.3 | 37.3 | 46.3 |
| **Ours** | 72.5 | 67.6 | 58.7 | 49.3 | 41.2 | 28.1 | 15.4 | 57.9 | 38.5 | 47.5 |
| Abs. Improve | 1.3 | 1.4 | 1.8 | 2.1 | 1.1 | 0.5 | 0.6 | 1.5 | 1.2 | 1.2 |
| DDG-Net(ICCV'23)[43] | 72.5 | 67.7 | 58.2 | 49.0 | 41.4 | 27.6 | 14.8 | 57.8 | 38.2 | 47.3 |
| DDG-Net$^\dagger$(ICCV'23)[43] | 71.2 | 66.3 | 57.9 | 48.5 | 40.6 | 27.6 | 15.3 | 56.9 | 37.9 | 46.7 |
| **Ours** | 73.0 | 69.2 | 59.6 | 50.2 | 41.8 | 28.4 | 14.9 | 58.8 | 39.0 | 48.2 |
| Abs. Improve | 1.9 | 2.9 | 1.8 | 1.7 | 1.2 | 0.9 | -0.3 | 1.9 | 1.1 | 1.5 |

**Table 2: The proposed methods are applied to different WTAL methods on ActivityNet-1.2 dataset.**

| Method | mAP@IoU(%) | | | AVG |
|---|---|---|---|---|
| | 0.5 | 0.75 | 0.95 | (0.5:0.95) |
| $CO_2$-Net[10] | 43.3 | 26.3 | 5.2 | 26.4 |
| $CO_2$-Net$^\dagger$[10] | 43.3 | 26.1 | 5.3 | 26.3 |
| **Ours** | 43.4 | 26.2 | 5.3 | 26.4 |
| Abs. Improve | 0.1 | 0.1 | 0.0 | 0.1 |
| DDG-Net[43] | 44.3 | 26.9 | 5.5 | 27.0 |
| DDG-Net$^\dagger$[43] | 43.5 | 26.2 | 5.5 | 26.6 |
| **Ours** | 44.4 | 27.0 | 5.5 | 27.1 |
| Abs. Improve | 0.1 | 0.8 | 0.0 | 0.5 |

### 4.2 Implementation Details

**Evaluation Metrics.** We follow the standard evaluation protocol by reporting mean average precision (mAP) values. A prediction proposal is considered correct if the Intersection over Union (IoU) between the predicted proposal and the ground truth is deemed satisfactory. For evaluation purposes, we employ the code directly provided by the baseline WTAL models [10, 3, 43]. In subsequent experiments, $\dagger$ denotes results obtained on our platform using public code provided by the authors.

**Feature Extractor.** Following previous work[33, 5, 10, 3, 43], the optical flow maps are generated using the TV-L1 algorithm [50], and we employ I3D network [1] pre-trained on the Kinetics dataset [15] to extract both RGB and FLOW features without fine-turning. Each video is segmented into 16-frame intervals following the settings in [10, 3, 43].

**Training Settings.** For a fair comparison, we utilize the same training hyperparameters as those used by the baseline WTAL models [10, 3, 43]. For THUMOS14 dataset, we set $\delta = 0.7$, $\theta_{rep} = 0.8$, $\theta_{pro} = 0.55$, $\beta = \gamma = 0.5$ when using $CO_2$-Net [10] as baseline model, $\delta = 0.8$, $\theta_{rep} = 0.9$, $\theta_{pro} = 0.6$, $\beta = \gamma = 0.5$ when using

DELU [3] as baseline model, $\delta = 0.9$, $\theta_{rep} = 0.75$, $\theta_{pro} = 0.86$, $\beta = \gamma = 0.6$ when using DDG-Net [43] as baseline model. Our method is implemented using PyTorch 2.2 and trained under Ubuntu Server 22.04 platform with single NVIDIA RTX 4070Ti GPU.

### 4.3 Effectiveness on Different Baselines

To demonstrate the effectiveness and generalizability of our proposed methods, we deploy our methods on three different WTAL models on THUMOS14 dataset, which are $CO_2$-Net [10], DELU [3] and DDG-Net [43]. The results are shown in Table 1, where we can see that our methods consistently outperform the baseline methods across average evaluation metrics. Specifically, the proposed method achieves an absolute improvement of 2.2%, 1.2% and 1.5% in terms of average mAP@IoU(0.1:0.7) over $CO_2$-Net, DELU and DDG-Net respectively. Notably, the improvement at IoU = 0.7 for DDG-Net is minimal, potentially due to DDG-Net's focus on modeling ambiguous snippets, whereas our methods effectively reduce the impact of confounding from such snippets, which may slightly decrease performance.

Table 2 presents the results on ActivityNet-1.2 Datasets for different WTAL methods. Compared with the baseline methods, our method still maintains a certain performance advantage, achieving an absolute improvement of 0.1% and 0.5% in average mAP@IoU(0.5:0.95) over $CO_2$-Net and DDG-Net respectively.

Experiments on both THUMOS14 and ActivityNet datasets confirm that our dual-confounding eliminating framework could realize $P(Y|do(V))$ estimation and improve the performance of WTAL methods. The improvement is achieved on different WTAL models, illustrating the effectiveness and generalizability of our method.

### 4.4 Comparisons with State-of-the-art Methods

We choose DDG-Net[43] as the baseline method to compare with state-of-the-art methods on THUMOS14 dataset. The comparison results are shown in Table 3. This table shows that our method outperforms the baseline method and achieves competitive performance compared with the state-of-the-art methods. Specifically,

**Table 3: Comparison results with existing methods on THUMOS14 dataset.**

| Supervision | Method | mAP@IoU(%) | | | | | | | AVG | | |
|---|---|---|---|---|---|---|---|---|---|---|---|
| | | 0.1 | 0.2 | 0.3 | 0.4 | 0.5 | 0.6 | 0.7 | (0.1:0.5) | (0.3:0.7) | (0.1:0.7) |
| Fully | SSN (ICCV'17) [63] | 60.3 | 56.2 | 50.6 | 40.8 | 29.1 | - | - | 49.6 | - | - |
| | BSN (ECCV'18) [23] | - | - | 53.5 | 45.0 | 36.9 | 28.4 | 20.0 | - | 36.8 | - |
| | GTAN (CVPR'19) [28] | 69.1 | 63.7 | 57.8 | 47.2 | 38.8 | - | - | 55.3 | - | - |
| | TRA (TIP'22) [62] | 73.7 | 72.6 | 70.0 | 64.3 | 57.4 | 46.2 | 31.1 | 67.6 | 53.8 | 59.3 |
| Weakly | BaS-Net(AAAI'20) [18] | 58.2 | 52.3 | 44.6 | 36.0 | 27.0 | 18.6 | 10.4 | 43.6 | 27.3 | 35.3 |
| | DGAM(CVPR'20) [39] | 60.0 | 54.2 | 46.8 | 38.2 | 28.8 | 19.8 | 11.4 | 45.6 | 29.0 | 37.0 |
| | FAC-Net (ICCV'21) [11] | 67.6 | 62.1 | 52.6 | 44.3 | 33.4 | 22.5 | 12.7 | 52.0 | 33.1 | 42.2 |
| | $CO_2$-Net(MM'21) [10] | 70.1 | 63.6 | 54.5 | 45.7 | 38.3 | 26.4 | 13.4 | 54.4 | 35.7 | 44.6 |
| | FTCL(CVPR'22) [6] | 69.6 | 63.4 | 55.2 | 45.2 | 35.6 | 23.7 | 12.2 | 53.8 | 34.4 | 43.6 |
| | DCC(CVPR'22) [20] | 69.0 | 63.8 | 55.9 | 45.9 | 35.7 | 24.3 | 13.7 | 54.1 | 35.1 | 44.0 |
| | RSKP(CVPR'22) [12] | 71.3 | 65.3 | 55.8 | 47.5 | 38.2 | 25.4 | 12.5 | 55.6 | 35.9 | 45.1 |
| | ASM-Loc(CVPR'22) [8] | 71.2 | 65.5 | 57.1 | 46.8 | 36.6 | 25.2 | 13.4 | 55.4 | 35.8 | 45.1 |
| | DGCNN (MM'22) [40] | 66.3 | 59.9 | 52.3 | 43.2 | 32.8 | 22.1 | 13.1 | 50.9 | 32.7 | 41.3 |
| | Li et al.(MM'22) [21] | 69.7 | 64.5 | 58.1 | 49.9 | 39.6 | 27.3 | 14.2 | 56.4 | 37.8 | 46.1 |
| | DELU(ECCV'22) [3] | 71.5 | 66.2 | 56.5 | 47.7 | 40.5 | 27.2 | 15.3 | 56.5 | 37.4 | 46.4 |
| | TFE-DCN(WACV'23) [64] | 72.3 | 66.5 | 58.6 | 49.5 | 40.7 | 27.1 | 13.7 | 57.5 | 37.9 | 46.9 |
| | Wang et al.(CVPR'23) [49] | 73.0 | 68.2 | 60.0 | 47.9 | 37.1 | 24.4 | 12.7 | 57.2 | 36.4 | 46.2 |
| | Li et al.(CVPR'23) [19] | - | - | 56.2 | 47.8 | 39.3 | 27.5 | 15.2 | - | 37.2 | - |
| | P-MIL(CVPR'23) [36] | 71.8 | 67.5 | 58.9 | 49.0 | 40.0 | 27.1 | 15.1 | 57.4 | 38.0 | 47.0 |
| | CASE(ICCV'23) [24] | 72.3 | - | 59.2 | - | 37.7 | - | 13.7 | - | - | - |
| | Wang et al.(ICCV'23) [46] | 75.1 | 68.9 | 60.2 | 48.9 | 38.3 | 26.8 | 14.7 | 58.3 | 37.8 | 47.2 |
| | DDG-Net(ICCV'23)[43] | 71.2 | 66.3 | 57.9 | 48.5 | 40.6 | 27.6 | 15.3 | 56.9 | 37.9 | 46.7 |
| | Baseline (DDG-Net$^{\dagger}$) | 71.2 | 66.3 | 57.9 | 48.5 | 40.6 | 27.6 | 15.3 | 56.9 | 37.9 | 46.7 |
| | **Ours** | 73.0 | 69.2 | 59.6 | 50.2 | 41.8 | 28.4 | 14.9 | **58.8** | **39.0** | **48.2** |

**Table 4: Comparison results with existing methods on ActivityNet-1.2 dataset.**

| | Method | mAP@IoU(%) | | | AVG |
|---|---|---|---|---|---|
| | | 0.5 | 0.75 | 0.95 | (0.5:0.95) |
| Fully | TAL-Net (CVPR'17)[2] | 38.2 | 18.3 | 1.3 | 20.2 |
| | BSN(ECCV'18) [23] | 46.5 | 30.0 | 8.0 | 30.0 |
| | GTAN(CVPR'19)[28] | 52.6 | 34.1 | 8.9 | 34.3 |
| Weakly | $CO_2$-Net(MM'21)[10] | 43.3 | 26.3 | 5.2 | 26.4 |
| | DELU(ECCV'22) [3] | 44.2 | 26.7 | 5.4 | 26.9 |
| | DELU(ECCV'22)$^{\dagger}$ [3] | 43.9 | 25.7 | 5.4 | 26.4 |
| | ACGNet(AAAI'22)[57] | 41.8 | 26.0 | 5.9 | 26.1 |
| | DGCNN(MM'22)[40] | 42.0 | 25.8 | **6.0** | 26.2 |
| | Li et al.(MM'22) [21] | 41.6 | 24.8 | 5.4 | 25.2 |
| | P-MIL(CVPR'23) [36] | 44.2 | 26.1 | 5.3 | 26.5 |
| | DDG-Net(ICCV'23)[43] | 44.3 | 26.9 | 5.5 | 27.0 |
| | Baseline (DDG-Net$^{\dagger}$) | 43.5 | 26.2 | 5.5 | 26.6 |
| | Ours | **44.4** | **27.0** | 5.5 | **27.1** |

our method achieves an absolute improvement of 1.5% in terms of average mAP@IoU(0.1:0.7) over the baseline method, and mAP has a significant improvement of 2.9% at the IoU of 0.2 without introducing additional learnable parameters and model re-designs. What's more, compared with some fully supervised methods, our method also achieves competitive performance, which demonstrates the advantages of our method in weakly supervised temporal action localization.

Table 4 presents the comparison results on ActivityNet dataset. It can be observed that our method achieves an absolute improvement of 0.5% in terms of average mAP@IoU(0.5:0.95) over the baseline method, and mAP has a significant improvement of 0.8% at the IoU of 0.75. The results demonstrate the effectiveness and generalizability of our method on different datasets and WTAL methods.

## 4.5 Ablation Study and Analysis

**Effects of components.** To investigate the effectiveness of each component in the proposed method, we conduct ablation studies on the THUMOS14 dataset with different WTAL methods. The results are shown in Fig. 5. For different baseline methods, SCM has a significant improvement in AVG mAP@IoU(0.1:0.7). Specifically, an absolute improvement of 1.7%, 0.8% and 1.1% in terms of AVG mAP@IoU(0.1:0.7) is achieved over $CO_2$-Net, DELU and DDG-Net respectively. These improvements demonstrate the SCM is effective in reducing the impact of confounding bias and improving the performance of WTAL methods. While only using MCM does not have a significant improvement in AVG mAP@IoU(0.1:0.7) over the baseline methods. However, when combining SCS and MCM, the proposed method achieves the best performance.

**Analysis of insight.** Reviewing our methods, the dual-confounding eliminating framework is proposed to approximate

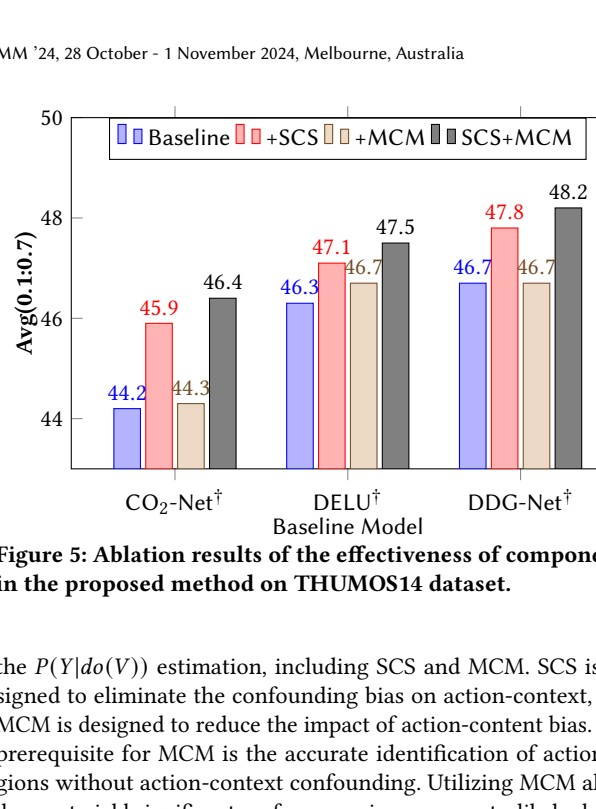

**Figure 5: Ablation results of the effectiveness of components in the proposed method on THUMOS14 dataset.**

the $P(Y|do(V))$ estimation, including SCS and MCM. SCS is designed to eliminate the confounding bias on action-context, and MCM is designed to reduce the impact of action-content bias. The prerequisite for MCM is the accurate identification of action regions without action-context confounding. Utilizing MCM alone does not yield significant performance improvements, likely due to the persistent confounding bias in action-context. If this bias is not effectively eliminated, MCM's effectiveness is compromised, resulting in negligible performance gains. However, with SCS effectively eliminating action-context confounding, the more accurate action regions are obtained. This improvement is beneficial for MCM, as it enables the enforcement of consistency between the discriminative snippets and their corresponding action instances. The synergistic combination of SCS and MCM achieves the optimal performance, demonstrating the effectiveness of our dual-confounding eliminating framework.

**Construction strategy of $\Psi$.** $\Psi$ is constructed based on the modal gap between RGB and FLOW features. Despite Eq. 2, we also try others construction strategies, such as

$$\Psi_i^{thre} = Index(\mathcal{A}_i^{RGB} > \xi \, \& \, \mathcal{A}_i^{FLOW} < \xi)$$
$$+ Index(\mathcal{A}_i^{RGB} < \xi \, \& \, \mathcal{A}_i^{FLOW} > \xi) \qquad (14)$$

where $\xi$ is a threshold, the "+" represents the concatenation of two index lists. Also, we try to use the absolute difference and the ratio of the two modalities, which are defined as

$$\Psi_i^{delta} = abs(\mathcal{A}_i^{RGB} - \mathcal{A}_i^{FLOW}) \qquad (15)$$

Besides, we also try to use the ratio of the two modalities, which is formulated as

$$\Psi_i^{minmax} = min(\mathcal{A}_i^{RGB}, \mathcal{A}_i^{FLOW})/max(\mathcal{A}_i^{RGB}, \mathcal{A}_i^{FLOW}) \qquad (16)$$

We conduct experiments to analyze the effects of different construction strategies of $\Psi$. The results are shown in Table 5. It can be observed that the construction strategy of $\Psi$ has a significant impact on the performance of our method. The proposed construction strategy in Eq. 2 achieves the best performance, which may be due to the fact that Eq. 2 could limit the values of $\Psi$ to [0, 1], which is beneficial for the subsequent causal intervention.

**Table 5: Ablation results on the construction strategy of $\Psi$. DDG-Net[43] is used as the baseline method.**

| Measurement | mAP@IoU(%) | | | AVG | |
|---|---|---|---|---|---|
| | 0.3 | 0.5 | 0.7 | (0.3:0.7) | (0.1:0.7) |
| Baseline (without $\Psi$) | 57.9 | 40.6 | 15.3 | 37.9 | 46.7 |
| $\Psi_i^{thre}$ (Eq. 14) | 59.2 | 42.0 | 15.1 | 38.3 | 47.8 |
| $\Psi_i^{delta}$ (Eq. 15) | 58.5 | 41.3 | 14.2 | 37.9 | 47.3 |
| $\Psi_i^{minmax}$ (Eq. 16) | 59.1 | 41.0 | 14.7 | 38.1 | 47.6 |
| $\Psi_i$ (Eq. 2) | 59.6 | 41.8 | 14.9 | **39.0** | **48.2** |

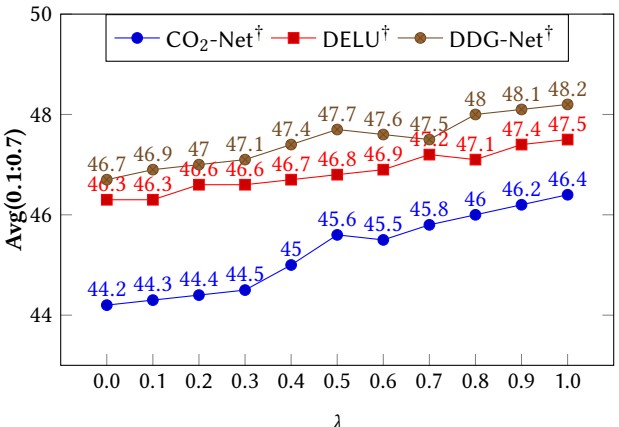

**Figure 6: Ablation results of $\lambda$ in the proposed method on THUMOS14 dataset.**

**Strength of Causal Intervention.** $\lambda$ represents the strength of causal intervention. We conduct experiments to analyze the effects of $\lambda$ on the performance of our method. The results are shown in Fig. 6. It can be observed that the performance of our method is sensitive to $\lambda$. Low $\lambda$ may lead to a weak causal intervention, which may not eliminate the confounding bias effectively. Overall, with the increase of $\lambda$, we observe an enhancement in overall performance, indicating that our SCS serves as an effective substitute confounder.

## 5 CONCLUSION

In this paper, we revisit WTAL from the perspective of structural causal model, and propose a dual-confounding eliminating framework. Specifically, we construct a substitute confounder set to eliminate the action-context confounding bias by leveraging the modal gap between RGB and FLOW features, and enforce the consistency between the discriminative snippets and the corresponding action instances to reduce the impact of action-content bias. Our methods can be seamlessly integrated into any two-stream model without the addition of parameters, using the EM algorithm, at a low computational cost. Extensive experiments on the THUMOS14 and ActivityNet-1.2 datasets demonstrate the effectiveness and generalizability of our methods. Through detailed ablation studies and analysis, the critical role of each component within our framework is highlighted, underscoring their collective contribution to overall performance.

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
