# OpenReview forum: "Efficient Dual-Confounding Eliminating for Weakly-supervised Temporal Action Localization"
_acmmm.org/ACMMM/2024/Conference — MM2024 Poster_

### Official Review · Reviewer_TsCG · 2024-05-22

**Rating:** 5
**Confidence:** 4

**Summary:**

This paper revisits WTAL from the perspective of structural causal model to identify the true origins of confounding, and propose an efficient dual-confounding eliminating framework to alleviate these biases. Specifically, a Substituted Confounder Set (SCS) is developed to eliminate confounding bias in action-content by exploiting the modal disparity between RGB and FLOW. Additionally, a Multi-level Consistency Mining (MCM) method is designed to reduce confounding bias in action-content by leveraging the consistency between discriminative snippets and their corresponding proposals at both the feature and label levels. Extensive experiments on two popular benchmarks demonstrate the effectiveness of the proposed method.

**Strengths:**

1. As far as I'm concerned, few existing methods have considered to adopt structural causal models to identify the origins of confounding. This should be a valuable direction for future research.

2. The paper is well-written and easy to understand. The model performance is strong. Extensive experiments demonstrate the effectiveness of the method. I like this paper generally.

3. I appreciate that the authors include the source code in the supplementary material.

**Limitations:**

1. I am curious that whether the model performance is sensitive to the hyperparameters $\beta$ and $\gamma$ in Eq.(13). An experiment of parameter analysis may help readers better understand the method.

**Suitability:**

3

---

### Official Review · Reviewer_sCaY · 2024-05-24

**Rating:** 4
**Confidence:** 2

**Summary:**

In this paper, the authors revisit Weakly-supervised Temporal Action Localization~(WTAL) from the perspective of the structural causal model to identify the true origins of confounding and propose an efficient dual-confounding eliminating framework to alleviate these biases. Specifically, a Substituted Confounder Set (SCS) is constructed to eliminate the confounding bias on action-content by leveraging the modal disparity between RGB and FLOW. A Multi-level Consistency Mining (MCM) method is designed to mitigate the confounding bias on action-content by utilizing the consistency between discriminative snippets and corresponding proposals at both the feature and label levels. Extensive experiments demonstrate the superior performance of the method.

**Strengths:**

1. Well written;
2. Detailed ablation studies.

**Limitations:**

1. The authors claim in Fig. 1~(a) and~(b) that the previous method exists Visual confounding and Motion confounding, how can the above problems be seen from the visual similarity? In addition, why Fig. 1. does not provide the results of this paper's method to show effectiveness. The same doubt exists for Fig. 2.
2. Why didn't this work compare Sota results on other dataset benchmarks, e.g. FinaAction, FineGym?
3. High-quality visualization results are missing in the experimental part to further demonstrate the advantages of the method.
4. What is the relation between SCS & MCM and the structural causal model?
5. In the SOTA comparison, do all the methods use both RGB and optical flow modalities?

**Suitability:**

2

---

### Official Review · Reviewer_VMhQ · 2024-05-25

**Rating:** 4
**Confidence:** 4

**Summary:**

Guided by Causal Intervention, this paper propose an efficient dualconfounding eliminating framework to mitigate the confounder on action-context and action-content.

**Strengths:**

This work revisit WTAL from the perspective of structural causal model to identify the true origins of confounding, and propose an efficient dual-confounding eliminating framework to alleviate these biases. Experiments on 2 challenging benchmarks including THUMOS14 and ActivityNet-1.2 demonstrate the superior performance of our method.

**Limitations:**

1. Is Eq.3 preprocessed or used at each epoch.

2. The proposed context regularization method is limited by dual streams.

3. In my views，the confounding action context and content are also accumulated by different action instances. Therefore, the action instance also seems to determine the action priors.

4. In Eq.3, the differences between RGB and OF are used to measure the confounder.  However, when the samples of both modalities are simultaneously confounding？ The difference doesn't seem to exceed the threshold either.

In summary, the method in this paper looks like a simple regularization of context and content. But the author utilizes causal intervention to endow regularity with interpretability. And this manuscript is reasonable and well-organized.

**Suitability:**

2

---

### Official Review · Reviewer_amFj · 2024-05-25

**Rating:** 4
**Confidence:** 3

**Summary:**

The authors revisit Weakly-supervised Temporal Action Localization (WTAL) from the perspective of the structural causal model to identify the true origins of confounding and propose an efficient dual-confounding eliminating framework to alleviate these biases. This framework consists of Substituted Confounder Set (SCS) and Multi-level Consistency Mining (MCM). The former is designed to eliminate the confounding bias on action-content by leveraging the modal disparity between RGB and FLOW. The latter is responsible for mitigating the confounding bias on action-content by utilizing the consistency between discriminative snippets and corresponding proposals at both the feature and label levels.

**Strengths:**

1. The authors revisit WTAL from the perspective of the structural causal model to identify the true origins of confounding.
2. The motivation of this paper is clear and novel.
3. Extensive experiments on THUMOS14 and ActivityNet-1.2 demonstrate the superior performance of the proposed method.

**Limitations:**

1. The term 'dual-confounding' is ambiguous; it is unclear whether it denotes two distinct solutions or refers to two cofounders.
2. Could you please specify which section of Figure 3 corresponds to the I element depicted in Figure 4?
3. For enhanced comprehension, it would be beneficial if the concept of 'cutting off C->V' is clearly illustrated in Figure 3.

**Suitability:**

3

---

### Meta-Review · Area_Chair_gMoi · 2024-07-04

**Recommendation:** Accept (Poster)
**Confidence:** 4

**Metareview:**

This paper revisits WTAL from the perspective of structural causal model to identify the true origins of confounding, and propose an efficient dual-confounding eliminating framework to alleviate these biases. All reviewers are satisfied with the rebuttal and unanimously recommend acceptance of the paper.

---

### Meta-Review · Senior_Area_Chairs · 2024-07-10

**Recommendation:** Accept (Poster)
**Confidence:** 4

**Metareview:**

All the reviewers gave positive ratings and tend to accept the paper. SAC and AC agree with reviewers and recommend accptance of the paper.